# Genomic Signatures of MASLD: How Genomics Is Redefining Our Understanding of Metabolic Liver Disease

**DOI:** 10.3390/ijms262210881

**Published:** 2025-11-10

**Authors:** Peter Saliba-Gustafsson, Jennifer Härdfeldt, Matteo Pedrelli, Paolo Parini

**Affiliations:** 1Cardio Metabolic Unit, Department of Medicine, Karolinska Institutet, 171 77 Stockholm, Sweden; 2Medicine Unit of Endocrinology, Theme Inflammation and Ageing, Karolinska University Hospital C2:94, 141 86 Stockholm, Sweden; 3Cardio Metabolic Unit, Department of Laboratory Medicine, Karolinska Institutet, 171 77 Stockholm, Sweden

**Keywords:** MASLD, genetics, functional genomics, GWAS, genome-wide association study, fatty liver disease, network medicine, perturb-seq

## Abstract

Metabolic dysfunction-associated steatotic liver disease (MASLD) is the most prevalent chronic liver condition globally, driven by strong genetic and environmental components. This review summarizes recent advances in understanding the genetic architecture of MASLD. Genome-wide association studies (GWAS) have identified several key risk variants, primarily in genes such as *PNPLA3*, *TM6SF2*, *GCKR*, and *MBOAT7*, which influence hepatic lipid metabolism and disease progression. By utilizing surrogate markers of MASLD, researchers have also identified numerous putative MASLD-associated genes, warranting further investigation through functional genomics approaches. Next-generation sequencing techniques have uncovered rare variants in genes like *APOB* and *ABCB4*, as well as protective variants in *HSD17B13* and *CIDEB*. This review discusses the potential of polygenic risk scores for disease stratification and the development of genetically informed therapeutic strategies. Additionally, it explores the future of functional genomics approaches in discovering novel treatment strategies. While the evolving genetic landscape of MASLD provides promising insights for precision medicine approaches in diagnosis, prognosis, and treatment, significant translational gaps remain. Addressing these challenges will be critical for realizing the full potential of personalised approaches in clinical management. This review synthesizes these findings and discusses their implications for future research and clinical practice in MASLD.

## 1. Introduction

Metabolic dysfunction-associated steatotic liver disease (MASLD), formerly known as non-alcoholic fatty liver disease (NAFLD), has risen to become the most common chronic liver condition across the globe. This disease is marked by excessive fat accumulation in the liver, affecting individuals with minimal or no alcohol consumption. It is closely intertwined with the global epidemics of obesity, type 2 diabetes, and metabolic syndrome [1,2,3]. The reach of MASLD is vast, with an estimated 25–30% of the adult population affected worldwide, though the exact prevalence varies based on region and ethnicity. In many Western countries, the rates soar even higher, with prevalence climbing to 30–40% [1,4,5].

MASLD does not merely result in liver fat accumulation; it can progressively worsen, potentially leading to more severe conditions such as metabolic dysfunction-associated steatohepatitis (MASH), liver fibrosis, cirrhosis, and, in some cases, hepatocellular carcinoma [1,2,6,7]. Moreover, the impact of MASLD extends far beyond the liver itself, affecting multiple organ systems and overall health. Patients with MASLD face a heightened risk of cardiovascular disease, type 2 diabetes, and chronic kidney disease [1,2,8,9].

The economic burden of MASLD is substantial, imposing significant costs on healthcare systems worldwide. In the United States alone, annual medical costs related to MASLD were estimated to be approximately $103 billion in 2016. With MASLD prevalence rising and more cases progressing to severe stages, the economic toll is poised to escalate even further [10,11].

## 2. The Environmental Catalysts of MASH

The modern sedentary lifestyle, combined with overconsumption of energy-dense and processed foods [12], drives a chronic positive energy balance, which is a primary driver of the global obesity epidemic. This, in turn, has led to a sharp rise in associated conditions such as type 2 diabetes, metabolic syndrome, and MASLD. If progression is not hindered, MASLD may advance into MASH, whose hallmark is discernable inflammatory components. In the U.S., the estimated prevalence of MASH increases significantly with obesity and diabetes, rising from around 12% in the general middle-aged population to 22% in individuals with diabetes, and reaching up to 33% in those with obesity [13,14].

Globally, MASH is seen in 15% to 30% of those with obesity, and the figure rises to as high as 70% among individuals with morbid obesity [15]. Among MASH patients worldwide, between 31% and 89% are reported to have obesity, and 33% to 56% have diabetes [15].

Despite these patterns, not all individuals with obesity develop MASH, and intriguingly, some lean individuals do. This suggests that genetic factors, in conjunction with environmental influences, play a significant role in the development of the disease [3].

## 3. Genetic Chronicles of MASLD: The Historical Exploration of Predisposition

The genetic predisposition to MASLD has been a key focus of research, with the *PNPLA3* [patatin like phospholipase domain containing 3] locus standing out as one of the most extensively studied. The *PNPLA3* rs738409 C>G single nucleotide polymorphism (SNP), which encodes the I148M variant, has emerged as the most consistently replicated genetic marker associated with heightened liver fat accumulation and an increased risk of MASLD [16]. This variant’s impact on hepatic fat content has been robustly demonstrated through various imaging modalities, including ultrasound and advanced MRI techniques. Carriers of the *PNPLA3* I148M variant exhibit significantly higher liver fat content than non-carriers, with homozygous individuals (GG genotype) having up to 73% more liver fat than those with the CC genotype [6,17,18].

The frequency of the *PNPLA3* risk allele differs across ethnic groups, providing a partial explanation for the observed variations in MASLD prevalence between populations. For example, the risk allele is more common in Hispanic populations, correlating with the higher rates of MASLD seen in these groups [3,19]. Environmental factors such as obesity and diet further amplify the risk, with obesity strengthening the association between the *PNPLA3* variant and liver fat accumulation [18,19,20,21,22,23,24,25].

Beyond its influence on steatosis, the *PNPLA3* I148M variant is also linked to more severe liver disease progression, including nonalcoholic steatohepatitis (NASH), fibrosis, and hepatocellular carcinoma [6,26,27,28,29,30,31,32].

*PNPLA3* belongs to a family of patatin-domain-containing lipid hydrolases that act on various substrates, including triacylglycerols, phospholipids, and retinol esters [33]. In humans, PNPLA3 is most highly expressed in hepatocytes and hepatic stellate cells, while in mice, it is predominantly found in adipocytes. Both human and mice also express PNPLA3 in the retina and other tissues [34]. PNPLA3 localizes to the surface of lipid droplets [35] and exhibits triglyceride lipase activity [36,37,38]. It is involved in lipid remodeling and the hepatic retention of polyunsaturated fatty acids [39,40]. Recent research indicates that PNPLA3’s enzymatic activity also facilitates the transfer of polyunsaturated fatty acids from triglycerides to phospholipids in hepatocytes [41], potentially affecting various aspects of hepatic lipid metabolism. Additionally, PNPLA3 demonstrates retinyl-palmitate lipase activity in vitro and participates in retinol release by hepatic stellate cells [42].

PNPLA3 expression is regulated by nutritional status. High carbohydrate levels increase PNPLA3 expression in mice [43,44] and human hepatocytes [45,46] through enhanced transcription and reduced protein turnover. Conversely, fasting decreases PNPLA3 levels [47]. Studies have shown that PNPLA3 plays a role in lipid metabolism. In rats on a high-fat diet, knockdown of wild-type *Pnpla3* reduced liver fat content by decreasing fatty acid esterification [48], which is consistent with in vitro studies demonstrating that Pnpla3 overexpression can promote lipogenesis in mammalian cells [49].

This understanding of genetic predisposition has opened avenues for more tailored interventions, suggesting that individuals carrying this risk allele may particularly benefit from lifestyle modifications [18,20,21,23,28].

While *PNPLA3* plays a prominent role, other genetic loci have also been implicated in MASLD. *TM6SF2* (Transmembrane 6 Superfamily Member 2) is another key gene, with the rs58542926 C>T variant linked to higher liver fat accumulation by influencing the secretion of very low-density lipoprotein (VLDL) from hepatocytes [50,51,52] as well as liver fibrosis [53]. TM6SF2 is localised to the endoplasmic reticulum (ER) and ER-Golgi intermediate compartment (ERGIC), where it facilitates the lipidation of nascent very low-density lipoproteins (VLDL), a critical step in lipoprotein maturation and secretion. The rs58542926 C>T (p.E167K) variant results in a misfolded TM6SF2 protein that undergoes accelerated degradation, impairing the lipidation process. Consequently, this leads to secretion of lipid-poor VLDL particles and intrahepatic triglyceride accumulation. Interestingly, despite reduced circulating lipids and apolipoprotein B levels, this variant paradoxically associates with increased MASLD severity and fibrosis progression. *TM6SF2* variants also influence plasma lipid profiles, contributing to reduced cardiovascular risk. Emerging evidence suggests *TM6SF2*’s involvement in modulating systemic inflammation, which may further impact disease progression [50,54]. This unexpected dissociation between hepatic steatosis and cardiovascular risk factors underlines the complex interplay between liver and extrahepatic tissues and between the impact on this interplay of the different genetic variants.

Another gene which seems to play an important role in MASLD is *MBOAT7* (Membrane Bound O-Acyltransferase Domain Containing 7), where the rs641738 C>T variant has been connected to elevated hepatic fat and inflammation. *MBOAT7* encodes a lysophosphatidylinositol acyltransferase involved in phosphatidylinositol remodeling, crucial for maintaining membrane lipid composition and cellular homeostasis. The rs641738 C>T variant diminishes *MBOAT7* expression, disrupting phospholipid remodeling and enhancing the susceptibility of hepatocytes to lipid accumulation and inflammatory activation [55]. This disturbance promotes aberrant lipid droplet formation and enhances pro-inflammatory signaling pathways, contributing to the progression from simple steatosis to steatohepatitis and fibrosis [56]. The inflammatory role of *MBOAT7* variants further implicates it in disease exacerbation distinct from purely metabolic mechanisms. Thus, variants affecting *MBOAT7* alter lipid droplet formation and inflammatory responses, further driving MASLD progression [57,58]. The association of the rs641738 C>T variant with CVD is less clear and somewhat controversial [58,59,60].

The *GCKR* gene (Glucokinase Regulatory Protein) also contributes to MASLD susceptibility. The rs1260326 C>T variant promotes hepatic glucose uptake and de novo lipogenesis, leading to increased liver fat [18,61,62].

Conversely, protective genetic variants have been identified, such as the *HSD17B13* rs72613567 T>TA variant, which offers a reduced risk of MASLD and its progression. This gene impacts lipid droplet formation and its nonfunctional variant appears to confer resistance against liver injury and fibrosis [63].

Other loci, such as *APOE* and *PPP1R3B*, offer intriguing insights into the metabolic complexity underlying MASLD. *APOE*, a central mediator of lipoprotein transport and cholesterol redistribution, shapes hepatic lipid homeostasis through its influence on remnant lipoprotein clearance, VLDL secretion, and triglyceride turnover. The differential effects of its allelic variants, notably *ε2* and *ε4*, underscore how subtle shifts in lipid trafficking can predispose to—or protect against—hepatic fat accumulation. In parallel, *PPP1R3B*, a key regulatory subunit guiding protein phosphatase 1 activity, governs hepatic glycogen synthesis and thereby impacts the delicate balance between carbohydrate storage and lipid deposition. Genetic variation at these loci not only reveals diverse mechanistic routes to steatosis but also points to an integrated metabolic network in which alterations in lipid and glucose handling converge to shape disease susceptibility and trajectory [18].

These discoveries offer invaluable insights into the genetic landscape of MASLD, aiding in the development of targeted treatment strategies. The impact of genetic loci on MASLD, lipid accumulation and liver fibrosis pathways is summarised in Figure 1. However, it is important to note that the effect sizes and directions of these genetic variants differ across ancestries, which impacts their generalisability and highlights the need for diverse population studies to ensure equitable clinical translation.

## 4. Unveiling the Hidden Genetic Risks of MASLD Through Surrogate Markers

Over the last decades, MASLD has been intensively studied with a reductionist approach that defined and classified the disease mainly according to the genetic variants found to be statistically associated with specific traits of the condition. Hence, the ample investment and application of genome-wide association studies (GWAS), with more than 5700 GWAS conducted in 10 years for more than 3300 diseases, has led to a remarkable range of discoveries in human genetics. Yet it has also contributed to the ‘one-gene, one-disease’ approach that fails to describe the complex and diverse clinical phenotypes characterizing MASLD. Standalone GWAS indeed do not consider the complex interaction between genes or environmental factors that significantly influence disease development and progression. A step forward has been achieved by combining MASLD genomics with various biomarkers as proxies for hepatic fat. For instance, leveraging alanine aminotransferase (ALT) levels as a surrogate for liver fat has identified notable genetic variants, including those in the glycerol-3-phosphate acyl-transferase (*GPAM*) and apolipoprotein E (*APOE*) genes, which modulate lipid synthesis and metabolism and are linked to MASLD risk [64,65]. Complementing this, magnetic resonance imaging (MRI)-based measurements in large biobanks have expanded the MASLD genetic landscape, revealing several loci such as *TRIB1*, *PNPLA2*, *APOH*, and *ADHB1*. Importantly, genes like *HFE* and *SERPINA1*, which are known to influence iron overload and endoplasmic reticulum stress, have maybe demonstrated a greater effect on cirrhosis than MASLD itself [66,67].

Further validation via computed tomography (CT) imaging and cross-validation with MRI highlighted new MASLD-related loci, including *FDGE5* and *CITED2* [67]. Meanwhile, a parallel GWAS using ALT levels in a multi-ancestry cohort revealed additional genetic hits related to insulin resistance, adiposity, and inflammation, with loci such as *PPARG*, *FTO*, *IL1R*, and *IFI30* coming to the fore [68].

Haas et al. took a novel approach by utilising MRI-derived liver fat measurements from the UK Biobank and applying machine learning to impute liver fat in subpopulations where MR imaging was available, but liver fat data was missing. This increased the sample size significantly, although few new associations were discovered, underscoring the challenges of MASLD GWAS due to MRI’s limited scalability [69].

Other strategies include the use of composite scores such as the fatty liver index (FLI). Li et al.’s GWAS using the FLI in the UK Biobank replicated known loci such as *PNPLA3* and *TM6SF2*, proving the utility of surrogates in expanding sample sizes for MASLD genetic studies [70]. However, the FLI has faced criticism for its limited predictive power, with waist-hip ratio shown to be equally effective in predicting MASLD [71].

Building on this, a novel MASLD-score was used to replicate historical loci and uncover new ones, highlighting the potential for expanding the genetics of MASLD through the use of surrogate markers [72].

While these surrogates offer promise for larger-scale genetic studies, caution must be exercised in interpreting the results, as non-specific phenotypic variables may confound associations. eQTL analysis in liver, and/or genetic colocalisation to liver tissue is essential to establish causal links before exploring the functional roles of SNPs identified through surrogate markers of MASLD.

## 5. The Polygenic Risk Score–Genetic Crystal Ball or Statistical Mirage?

A polygenic risk score (PRS) quantifies an individual’s genetic predisposition to develop a specific disease or experience a related outcome. It is calculated by summing the number of trait-associated alleles carried by an individual, often weighted by their effect size on the trait. These risk variants are identified through large-scale genome-wide association studies. When developing a PRS, several key factors must be clearly defined:The specific phenotype under investigationCharacteristics of the study population (e.g., risk status, ethnicity)The statistical model to be employedWhether non-genetic variables will be incorporated into the analysis

Careful consideration of these factors is crucial for creating an accurate and meaningful PRS [73].

PRS at least start to consider the polygenic nature of MASLD, thus reflecting the combined influence of multiple pathways rather than attributing risk to a single underlying mechanism. Hence PRS can be considered a less reductionistic approach to the disease than GWAS. However, it should be noted that these genetic approaches are particularly relevant in understanding rare genetic conditions associated with MASLD, such as variants in *PNPLA3*, *TM6SF2* or *HSD17B13*, which have been shown to influence disease severity. Yet, as with CVD, the association between a specific risk factor and MASLD in large populations does not necessarily demonstrate its causal implication. For instance, while certain genetic variants increase hepatic fat accumulation and fibrosis risk, they may not directly correlate with other metabolic comorbidities such as cardiovascular disease or diabetes in all cases.

PRSs have the potential to serve as non-invasive diagnostic tools for predicting long-term complications of fatty liver disease [74,75,76]. In fact, researchers have proposed that PRS-estimated FLD could be used as a proxy for diagnosis in high-risk individuals who exhibit known risk factors. This approach could provide valuable insights into disease progression and help identify patients who may benefit from early intervention or closer monitoring [77]. However, the primary value of PRSs may lie in their integration with clinical variables in routine healthcare. This combination could be used to identify individuals with fatty liver disease and/or metabolic disorders who are at higher risk of disease progression towards MASH and advanced fibrosis. Such patients would require closer monitoring and/or more intensive management. In this context, PRSs could also serve as valuable tools to predict individual responses to lifestyle changes or pharmacological interventions currently undergoing clinical trials. This approach could help tailor treatment strategies and improve patient outcomes by enabling more personalised and targeted interventions.

To further assess the clinical utility of polygenic risk scores (PRS), external validation metrics such as the area under the receiver operating characteristic curve (AUROC) and net reclassification improvement (NRI) have been increasingly used. These metrics are essential for determining the predictive accuracy of PRS in real-world, diverse populations. AUROC quantifies the discriminatory power of a model, indicating how well the PRS differentiates between individuals with and without the disease, while NRI assesses the reclassification of patients to more appropriate risk categories when using PRS alongside traditional risk factors. While these metrics have demonstrated potential in initial studies, their application in clinical settings remains limited, particularly for MASLD. The clinical implementation of PRS in predicting long-term disease outcomes would benefit from further external validation in diverse populations [75,78].

Equity limitations also pose a significant challenge in the widespread use of PRS. The vast majority of GWA studies that form the foundation for PRS have been conducted predominantly in individuals of European ancestry. This geographic and ethnic homogeneity creates a genetic diversity gap, which significantly limits the generalisability of PRS to non-European cohorts, particularly for populations of African, Asian, or Latin American descent. As a result, the clinical utility of PRS could be compromised in these populations, potentially exacerbating health disparities if not addressed. To ensure equitable healthcare implementation, future research must prioritize the inclusion of diverse ethnic groups in GWAS and develop population-specific PRS models. This will help to improve the accuracy and relevance of PRS for non-European populations and ensure that such tools do not inadvertently worsen existing health inequities [79].

With great opportunity comes also some limitations, and these must be acknowledged. The clinical application of PRS faces several significant limitations that warrant careful consideration before widespread implementation. A primary concern is the current lack of robust scientific evidence supporting their clinical utility. While studies have demonstrated the ability of PRS to improve the estimation of risk, there is a notable absence of evidence showing that their use leads to improved health outcomes [80,81]. This gap highlights the need for rigorous prospective studies and randomised controlled trials to establish the true value of PRS in practice. It also argues for moving beyond strict reductionism toward a systems-medicine approach that studies genetic variants within the context of their interactions across biological networks. Another critical limitation is the issue of equity in PRS application across diverse populations. The majority of genome-wide association studies used to develop PRS have been conducted primarily in populations of European ancestry. This bias significantly limits the generalisability and accuracy of PRS in other racial and ethnic groups, potentially exacerbating existing health disparities if implemented without addressing this limitation [82]. The use of PRS in clinical settings also raises concerns about potential unintended consequences on patient care and psychology. There are worries that providing patients with information about high genetic risk could lead to increased anxiety, depression, or fatalism. Conversely, low-risk scores might create a false sense of security, potentially leading to neglect of other important health behaviors [82]. Practical challenges in healthcare system implementation present another significant hurdle. These include uncertainties about how to effectively integrate PRS with other clinical risk factors, the lack of clear guidelines on using PRS to guide medical management, and potential issues with insurance coverage for PRS-based screening and interventions [82]. Lastly, the use of genetic information in clinical practice raises important ethical considerations. These include concerns about genetic privacy, the potential for discrimination based on genetic risk, and the complex issues surrounding the use of PRS for prenatal or preimplantation testing. These ethical dilemmas require careful consideration and the development of robust policies to protect patients’ rights and interests [82].

In conclusion, while PRS hold promise for personalised medicine, these limitations highlight the need for further research, policy development, and ethical guidelines before their widespread adoption in clinical practice. Their limitations in addressing the complexity of gene-gene and gene-environment interactions highlight the need for integrative approaches that better capture the multifactorial nature of this disease. Addressing these challenges is crucial to ensure that the implementation of PRS in healthcare is both scientifically sound and ethically responsible [80,81,82]. While PRS provide valuable risk stratification, increasingly, network medicine and integrative systems-level genomic approaches are being developed to unravel the complex molecular interplay underlying MASLD. These methods consider genetic variants within the context of extensive biological networks, offering insights beyond individual risk alleles and reflecting the multifactorial nature of the disease.

## 6. Dissecting MASLD Complexity: Network Medicine and Systems Genomics

Network analysis and its application to medicine—i.e., Network Medicine—views MASLD as a systems-level disorder arising from perturbations in interconnected molecular and cellular networks rather than from single genes or from its variants. By integrating genetic variants with transcriptomic, proteomic, and microbiome data, it maps disease modules and hubs that contextualize risk alleles and reveal pathways and targets across the spectrum from steatosis to fibrosis. These integrative multi-omics approaches are revolutionising our understanding of MASLD by transforming static genetic associations into dynamic biological narratives. For instance, disease-specific eQTL mapping has uncovered how the rs2291702 variant attenuates *AGXT2* expression, unveiling a previously hidden anti-fibrotic mechanism validated through rigorous experimental models [83]. Large-scale GWAS combined with eQTL and splicing QTL analyses have pinpointed causal genes at classic loci such as *PNPLA3*, *TM6SF2*, and *MBOAT7*, illuminating regulatory networks that orchestrate disease progression [65]. Meanwhile, protein interaction networks contextualize these genetic signals within lipid metabolism hubs, providing a rich framework to identify new therapeutic targets [84]. By decomposing polygenic risk into mechanistic axes, we now appreciate how distinct pathways—from lipoprotein secretion to fatty acid oxidation—define patient risk profiles. Furthermore, protective variants in genes like *HSD17B13* and *MTARC1* elegantly highlight nature’s own blueprint for resilience, offering promising avenues for intervention [63]. This synergy of genetics and systems biology is not just mapping MASLD; it is rewriting the roadmap toward precision medicine.

This evolving understanding of MASLD genetics, moving from large-scale association studies and risk prediction through PRS to network analyses’ roadmap to precision medicine and mechanistic insight, naturally leads us to explore functional genomics approaches. These methods enable direct interrogation of causal variants and the biological pathways they influence, thereby bridging the gap between genetic risk and molecular function.

## 7. Unraveling MASLD Mechanisms: Functional Genomics as a Gateway to Novel Therapeutic Targets

Building upon the insights gained from polygenic risk scores and genetic association studies, functional genomics provides an invaluable set of tools to directly identify and characterize the gene programs and causal mechanisms underlying MASLD. Through advanced techniques, such as eQTL mapping and CRISPR-based perturbation screens, researchers can translate genetic associations into actionable biological understanding.

Historically, the first step to inferring causality in human molecular genetic studies have been to assess (e)QTL effects of the disease-associated SNP. While this may provide some insight into the pathway or mechanism of action of the associated SNP, simple QTL analyses are too blunt a tool. Genetic colocalisation is a superior method to inferring causality, which is especially true for those methods that integrate fine-mapping and considering LD-structures in the locus to tease out the causal SNP [85]. One limitation to both simple QTL-analysis and colocalisation analyses in MASLD is the scarcity of publicly available RNA-seq data on liver samples. Further, care must be taken when considering what type of liver tissue is being analysed, e.g., healthy liver versus steatotic or fibrotic.

Revolutionising our understanding of complex diseases, CRISPR perturbation screens with transcriptomic readout, known as Perturb-seq, have emerged as a powerful tool in the field of genomics. This innovative approach systematically and unbiasedly illuminates the intricate connections between disease variants and their target genes, while simultaneously prioritising converging gene programs associated with risk for complex disorders [72,86,87,88]. Notably, we have pioneered the application of Perturb-seq specifically in MASLD by developing a HepaRG cell model system amenable to large-scale CRISPR interference screening and single-cell transcriptomics, as recently published in Hepatology [72]. This dataset provides MASLD-focused insights into causal gene networks and lipid metabolism regulation. Perturb-seq represents a quantum leap in our ability to decipher the genetic architecture of diseases. By combining the precision of CRISPR technology with the comprehensive insights of transcriptomics, this method offers an unprecedented view into the molecular underpinnings of health and disease. It allows researchers to:Unravel the complex web of genetic interactions that contribute to disease riskIdentify previously unknown gene targets associated with specific variantsPrioritize key gene programs that may serve as focal points for therapeutic intervention

While Perturb-seq offers exceptional resolution and mechanistic understanding, considerations such as data availability and reproducibility are essential, particularly when translating findings from cellular models to human pathology. That said, this groundbreaking approach is not just advancing our scientific knowledge; it’s paving the way for more targeted and effective treatments. By providing a clearer picture of how genetic variations influence cellular processes, Perturb-seq is helping to bridge the gap between genetic discoveries and clinical applications.

Recent advances in liver tissue engineering and functional genomics have underscored the potential of supramolecular chemistry-based methods in understanding and treating metabolic liver diseases like MASLD. Supramolecular approaches involve the self-assembly of molecules into complex, dynamic architectures through reversible, non-covalent interactions. This enables the creation of biologically compatible scaffolds that closely mimic the native liver microenvironment, which is essential for studying the mechanisms of MASLD and testing novel therapeutic strategies.

Supramolecular hydrogels and nanofibers, in particular, have emerged as versatile, carrier-free platforms that exhibit tunable mechanical properties and excellent biocompatibility. These scaffolds support three-dimensional (3D) hepatic cell cultures, promoting the maintenance of liver-specific functions over extended periods—thus overcoming the limitations of conventional 2D cultures and static tissue models. The use of such advanced biomaterials has allowed for more accurate modeling of key disease features, such as inflammation, fibrosis, and metabolic dysregulation, which are critical to understanding the molecular drivers of MASLD.

Recent studies have demonstrated how supramolecular assemblies can be leveraged to enhance liver tissue regeneration, improve functional maintenance, and better replicate the physiological conditions of MASLD [89,90]. These cutting-edge approaches offer new avenues for both basic research and translational applications in metabolic liver diseases.

## 8. Chronicling the Cure: MASLD and MASH Interventions

Translating genetic insights into effective therapeutic targets for MASLD and MASH has proven to be a complex endeavor [91]. At the forefront of this challenge are the influential genetic variants: the *PNPLA3* p.I148M missense variant and the *HSD17B13*-rs72613567 loss-of-function variant. While these variants may induce only subtle changes in gene and protein expression, they represent promising ‘druggable’ targets that could transform the treatment landscape for MASH. Since its identification in 2008 [16], the *PNPLA3* p.I148M variant has emerged as a key target for pharmacological intervention. Recent innovative strategies include:Liver-targeted GalNAc3-conjugated antisense oligonucleotide (ASO): This approach focuses on silencing *PNPLA3* in a knock-in mouse model, showcasing the potential for targeted therapy [92].Allele-specific siRNA: Tested in *PNPLA3* I148M-expressing mice, this strategy aims to mitigate MASH in diet-induced models [93].siRNA-lipid nanoparticles: By silencing *PNPLA3* p.I148M overexpression, researchers have successfully prevented the onset and progression of MASH in mice subjected to a high-fat Western diet [94].

These pioneering methods underscore the promise of precision medicine in treating MASLD.

A notable clinical trial (NCT04483947) is currently evaluating AZD2693, a ligand-conjugated antisense (LICA) drug designed to inhibit the production of PNPLA3 protein in participants carrying the 148M risk allele and diagnosed with MASLD/MASH.

In addition to antisense oligonucleotides (ASO) and small interfering RNA (siRNA) therapies targeting *PNPLA3*, emerging precision medicine approaches offer promising avenues for individualised MASLD treatment:Allele-specific modulation: Techniques that selectively modulate expression or function of the disease-associated allele, sparing the wild-type allele, are under investigation to minimize off-target effects and enhance therapeutic specificity. This includes allele-specific siRNA or antisense oligonucleotides designed to selectively silence the risk variant transcript [93].CRISPR-based gene editing: Genome editing platforms, including CRISPR-Cas9 and prime editing, have demonstrated the potential to correct pathogenic variants such as *PNPLA3* I148M in hepatocytes and organoid models, opening the path toward durable genetic cures. These approaches enable precise manipulation of genomic loci to restore normal gene function or disrupt harmful mutant alleles. Recent studies have successfully generated isogenic human hepatocyte organoids with edited *PNPLA3* alleles for mechanistic and drug screening applications [95].Small-molecule modulators: Identification and development of small molecules that modulate the activity of proteins encoded by genetic risk loci, or the downstream pathways they influence, are also advancing. Such compounds could complement RNAi or gene editing therapies by fine-tuning metabolic or inflammatory pathways implicated in MASLD progression [96].

These forward-looking strategies, alone or combined with existing RNA-based approaches, represent the frontier in tailoring MASLD therapeutics to patients’ individual genetic backgrounds, heralding a new era of precision hepatology.

The year 2024 has heralded a monumental breakthrough in the treatment of steatotic liver disease, marked by the U.S. FDA’s accelerated approval of Rezdiffra. This approval was based on the Phase 3 MAESTRO-NASH trial, a randomised, placebo-controlled study enrolling adults with biopsy-confirmed metabolic dysfunction-associated steatohepatitis (MASH) and moderate to advanced liver fibrosis (stages F2 to F3). This groundbreaking development, while not rooted in human genetics, represents a paradigm shift in our approach to MASH. Rezdiffra, a novel thyroid hormone receptor-beta (THR-β) agonist, emerges as a beacon of hope for adults grappling with noncirrhotic MASH accompanied by moderate to advanced liver fibrosis (stages F2 to F3). This innovative therapeutic, designed to complement diet and exercise, targets the very core of liver dysfunction. At the heart of Rezdiffra’s efficacy lies resmetirom, a partial agonist for THR-β, which is predominantly expressed in the liver. By activating THR-β, resmetirom orchestrates a reduction in intrahepatic triglycerides, addressing a key driver of MASH progression. The results of clinical trials have been nothing short of remarkable. MASH resolution without fibrosis worsening was achieved in 25.9% and 29.9% of patients receiving 80 mg and 100 mg of resmetirom, respectively, compared to a mere 9.7% in the placebo group. Furthermore, fibrosis improvement by at least one stage, without worsening of the NAFLD activity score, was observed in 24.2% (80 mg) and 25.9% (100 mg) of resmetirom-treated patients, surpassing the 14.2% seen in the placebo group. Beyond its primary endpoints, Rezdiffra demonstrated a profound impact on lipid profiles. Low-density lipoprotein cholesterol (LDL-C) levels decreased significantly from baseline to week 24: 13.6% in the 80 mg group and 16.3% in the 100 mg group, in stark contrast to the negligible 0.1% reduction observed in the placebo group [97]. Aramchol and its meglumine reformulation, developed by Galmed Pharmaceuticals, are oral SCD1 modulators targeting MASLD and MASH. Preclinical and clinical studies show effective SCD1 downregulation and antifibrotic effects. The Phase 3 ARMOR program—open-label (ARCON) 1-year results demonstrated histological fibrosis improvement and promising biomarker profiles. The main registrational study (NCT04104321) uses an optimised, higher-bioavailability meglumine form, supporting Aramchol’s positioning as a novel, oral therapy for MASLD/MASH fibrosis [98].

## 9. Conclusions

The horizon of MASLD research is poised for a revolutionary transformation. The widespread adoption of cutting-edge functional genomic technologies promises to unlock the secrets MASLD. By integrating functional evidence with MASLD-associated loci, we are not merely expanding our knowledge; we are opening gateways to groundbreaking therapeutic innovations. This synergy of genomics, functional biology and network medicine holds the key to unraveling novel treatment targets, paving way for truly personalised medicine in MASLD management. As we venture into this exciting frontier, we are not just treating a disease; we are transforming the future of liver health, tailored to each individual’s unique genetic blueprint. The potential for transformative breakthroughs in MASLD treatment is not just promising—it is within our grasp, heralding a new dawn in precision hepatology. We are entering a new, bright, and exciting future of MASLD functional genomics.

## Figures and Tables

**Figure 1 ijms-26-10881-f001:**
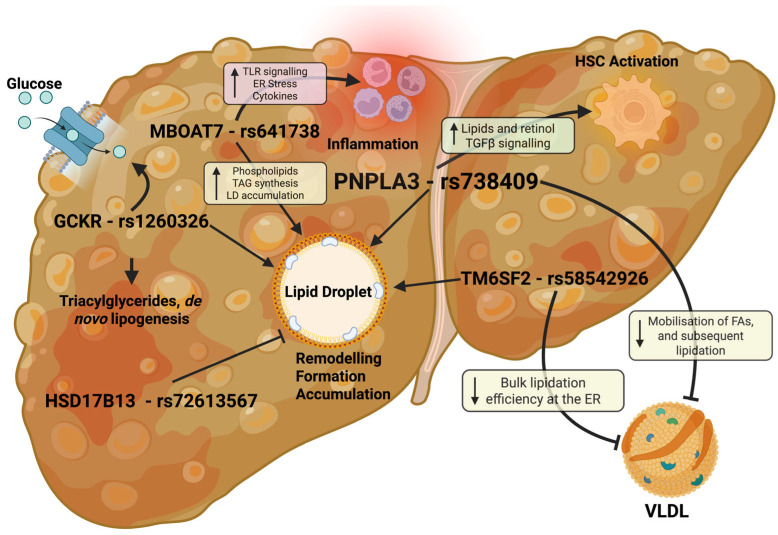
Summary of the influence of MASLD-associated genetic loci on liver biology and their suspected mechanisms of action. PNPLA3, the most extensively studied MASLD-associated locus, affects multiple pathways leading to increased hepatic lipid accumulation and MASLD progression. Notably, PNPLA3 regulates the mobilization of polyunsaturated fatty acids (PUFAs) from intracellular triglycerides to phospholipids, which facilitates the lipidation of ApoB-containing lipoproteins such as VLDL. The rs738409 variant impairs this process, reducing VLDL secretion and promoting lipid retention. Concurrently, altered lipid and retinol metabolism in hepatic stellate cells promotes their activation, amplified by enhanced TGF-β signaling. GCKR influences glucose uptake and de novo lipogenesis, increasing intracellular lipid storage. MBOAT7 deficiency caused by the rs641738 variant alters phosphatidylinositol remodeling, enhancing Toll-like receptor (TLR)-mediated inflammatory responses and mitochondrial dysfunction, while also promoting triglyceride synthesis and lipid droplet accumulation. TM6SF2 affects VLDL secretion by impairing triglyceride incorporation during VLDL lipidation in the endoplasmic reticulum and ER-Golgi intermediate compartment, leading to reduced secretion of large triglyceride-rich VLDL particles. HSD17B13 acts as a protective locus by reducing lipid accumulation in hepatocyte lipid droplets. Importantly, these MASLD-associated loci converge on the regulation of lipid droplet formation, remodeling, and lipid storage in hepatocytes. Figure key: yellow boxes indicate lipid accumulation/secretion pathways, green boxes represent stellate cell activation, and red boxes correspond to inflammatory pathways.

## Data Availability

No new data were created or analysed in this study.

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
