# Peer review of "Genomic Signatures of MASLD: How Genomics Is Redefining Our Understanding of Metabolic Liver Disease"

_ijms, 2025, doi:10.3390/ijms262210881_

Round 1
Reviewer 1 Report
Comments and Suggestions for Authors
This manuscript presents a comprehensive and timely review of the genetic and genomic landscape of Metabolic dysfunction-associated steatotic liver disease (MASLD). The authors effectively synthesize a vast body of literature, moving from well-established genetic risk variants (e.g., PNPLA3, TM6SF2) to more contemporary topics such as polygenic risk scores (PRS), functional genomics, and novel therapeutic strategies. The topic is of high relevance, given the global prevalence of MASLD and the accelerating pace of genetic discovery in this field. The review is generally well-structured and informative.
- While the title and abstract promise to explain how genomics is "redefining our understanding," the review often reads more as a summary of genetic associations rather than a synthesis of how these findings have fundamentally altered pathophysiological models. The conclusion hints at this, but the core argument could be more explicitly woven throughout the text.
- The organization, while comprehensive, could be streamlined. The transition from historical genes to surrogate markers is good, but the section "The Polygenic Risk Score" feels somewhat disconnected from the subsequent "Functional Genomics" section. The "Chronicling the Cure" section on interventions appears before the "Functional Genomics" section, which logically should underpin the discovery of those therapeutic targets.
- The section on functional genomics is promising but currently one of the briefest. Given its importance for the "redefining" theme and future directions, it deserves expansion.
- The authors mentioned metabolic liver disease, the supramolecular method should be introduced. To support this claim, the following important studies should be cited: Coord. Chem. Rev. 2024, 517, 216054; Adv. Mater. 2024, 36, 2304249.
Author Response
We sincerely thank the reviewer for their constructive and insightful feedback. In response, we have reorganized the manuscript to improve its overall flow and coherence. Specifically, we have expanded the Functional Genomics section to provide a more comprehensive and forward-looking discussion. Additionally, we have introduced a new section on supramolecular methods, incorporating the key studies recommended (Coord. Chem. Rev. 2024, 517, 216054; Adv. Mater. 2024, 36, 2304249) to enrich the mechanistic framework and highlight emerging approaches relevant to MASLD.
The revised manuscript with track changes for your review can be found as a "Supplementary file". The clean version of the manuscript is attached as "Manuscript file".
Reviewer 2 Report
Comments and Suggestions for Authors
This review provides a comprehensive overview of the genomic signatures underlying metabolic dysfunction-associated steatotic liver disease (MASLD) and systematically summarizes recent advances in genomics that have reshaped our understanding of the disease. It effectively covers key aspects, including the identification of genetic loci, the application of surrogate markers, the development of polygenic risk scores (PRS), and emerging therapeutic interventions, offering a valuable reference framework for both basic research and clinical translation in MASLD. However, several structural and content-related refinements are recommended to further enhance the manuscript’s logical flow and depth.
- The current sequence, where “Unveiling the Hidden Genetic Risks of MASLD through Surrogate Markers” follows “Genetic Chronicles of MASLD”, is conceptually sound, reflecting the transition from known genetic loci to potential undiscovered risks. However, in the chapter “Chronicling the Cure: MASLD and MASH Interventions,” the discussion begins with non-genetically related drugs (e.g., Rezdiffra) before moving to genetically informed therapeutic targets (e.g., PNPLA3, HSD17B13). This order may cause a logical inversion of “clinical intervention preceding mechanistic basis.” It is recommended to restructure the section to first discuss therapeutic targets derived from genetic mechanisms, followed by non-genetically related interventions, thereby strengthening the “mechanism to intervention” narrative.
- In “The Polygenic Risk Score – Genetic Crystal Ball or Statistical Mirage?” section, the discussion on “Network Medicine” represents an expanded systems-level perspective on genomic research rather than a direct analysis of PRS. To maintain thematic coherence, this content should either be developed into an independent section or integrated into a more suitable part of the manuscript addressing advanced or integrative genomic approaches.
- For key genes such as TM6SF2 and MBOAT7, additional details regarding the molecular pathways through which their variants contribute to disease progression should be included. For example, the interaction between TM6SF2 and proteins involved in VLDL assembly, or the role of MBOAT7 in modulating inflammatory signaling. This will ensure a more balanced and mechanistically consistent presentation across the genetic loci discussed.
- In the section on therapeutic interventions based on genetic targets, the manuscript currently focuses on PNPLA3-targeted approaches such as antisense oligonucleotides (ASO) and small interfering RNA (siRNA). To enhance translational relevance, the authors are encouraged to also discuss emerging precision medicine strategies, such as allele-specific modulation, CRISPR-based gene editing, or small-molecule modulators targeting genetic pathways, which represent promising avenues for individualized therapy.
- Figure 1 is too simplistic. Please supplement it with a "Schematic Diagram of the Mechanisms of Key Genes in MASLD" to intuitively illustrate the molecular pathways of genes such as PNPLA3, TM6SF2, and MBOAT7.
Author Response
We sincerely thank the reviewer for their valuable comments and thoughtful suggestions. We believe that we have sufficiently addressed all the concerns raised throughout the manuscript. Regarding the specific suggestion on Figure 1, we chose to address this in the text by providing detailed mechanistic descriptions rather than adding a more complex schematic figure. This decision was made to ensure the content remains accessible and engaging for a broad readership without overwhelming detail. We hope this approach provides a clear and balanced presentation of the key molecular pathways. We appreciate the reviewer’s understanding and hope this will be acceptable. We are confident that the revisions have strengthened the manuscript.
The revised manuscript with track changes for your review can be found as a "Supplementary file". The clean version of the manuscript is attached as "Manuscript file".
Reviewer 3 Report
Comments and Suggestions for Authors
Dear authors,
This review paper provides a clear and well-structured synthesis of the current genetic and functional genomic landscape of metabolic dysfunction–associated steatotic liver disease. The topic is timely and relevant, given the recent redefinition of NAFLD as MASLD and the growing interest in genetic determinants such as PNPLA3, TM6SF2, MBOAT7, HSD17B13, and GCKR.
The manuscript is generally well written, with fluent academic English and appropriate referencing. The graphical figure (page 4) effectively illustrates how key genetic loci contribute to lipid accumulation and inflammation pathways.
However, several structural and content-related issues need to be addressed before acceptance, particularly the missing author, consistency in terminology, and some overstatements regarding clinical applicability.
On the title page, the author list ends with “Paolo Parini 1,2 and …” with no name following “and,” which suggests the last author is missing. Please complete the author list and ensure all affiliations and the corresponding-author designation follow IJMS formatting.
Abstract & Introduction (p.1–2): The scope is clear, but consider tempering claims on “personalised medicine” with a brief qualifier about current translational gaps to avoid overpromising.
Genetic architecture overview (pp.3–5): Nicely synthesizes PNPLA3, TM6SF2, MBOAT7, GCKR, and protective HSD17B13; a one-sentence note that effect sizes and directions differ across ancestries would strengthen generalizability.
The genetic architecture section focuses on European and Hispanic populations but omits discussion of variability in other ethnic groups (e.g., East Asian, African). Including a sentence summarizing the current limitations in GWAS diversity would improve global relevance.
PRS section (pp.6–8): The discussion is balanced; consider adding a short paragraph on external validation metrics (AUROC, NRI) and equity limitations in non-European cohorts to guide clinicians.
The PRS section is comprehensive but lacks mention of validation metrics such as AUROC, NRI, or calibration statistics. It would be helpful to include at least one example of an externally validated MASLD PRS and to highlight the challenge of cross-ethnic transferability in these models.
Therapeutics (pp.8–9): The resmetirom Phase 3 summary is useful; adding study design details (biopsy-confirmed endpoints, F2–F3 inclusion) in one sentence would improve precision.
The discussion of CRISPR-based perturbation (Perturb-seq) is excellent, but it reads like a promotional summary. Please clarify whether any MASLD-specific Perturb-seq datasets currently exist or if the examples are extrapolated from other metabolic or cardiovascular diseases. A short paragraph on data availability and reproducibility would strengthen this section.
This is a well-composed and comprehensive narrative review on the genomics of MASLD. The major strengths lie in its integration of genetic mechanisms and functional genomic tools. However, the incomplete author information and overstatements regarding clinical translation warrant revision before publication.
Author Response
We sincerely thank the reviewer for their thorough and constructive feedback. We are pleased that the overall synthesis, writing quality, and relevance of the manuscript were well received.
In response to the specific concerns raised:
-
The author list on the title page has been carefully reviewed, we apologise the typo - there is no author missing, but just a misplaced ",".
-
The abstract and introduction have been revised to temper statements regarding personalized medicine by clearly acknowledging the current translational gaps, thereby avoiding overpromising.
-
In the genetic architecture overview, we have added sentences addressing the variability of effect sizes and directions across ancestries, as well as the limitations of GWAS diversity in non-European populations to improve global relevance.
-
The Polygenic Risk Score section now includes a short paragraph on external validation metrics (AUROC, NRI) and discusses equity limitations, including cross-ethnic transferability challenges, supported by examples of externally validated MASLD PRS models.
-
We have expanded the therapeutics section by adding study design details for the resmetirom Phase 3 trial, including biopsy-confirmed endpoints and F2–F3 fibrosis stages, to improve precision.
-
The Functional Genomics section has been enhanced by clarifying that our group has generated MASLD-specific Perturb-seq datasets (Saliba-Gustafsson et al., Hepatology 2024), providing unique insights into causal gene networks. We also included a discussion on data availability and reproducibility, balancing enthusiasm with realistic appraisal.
We believe these revisions effectively address all the reviewer’s concerns and significantly strengthen the manuscript’s clarity, depth, and translational relevance. We appreciate the reviewer’s careful consideration and helpful suggestions.
The revised manuscript with track changes for your review can be found as a "Supplementary file". The clean version of the manuscript is attached as "Manuscript file".
Round 2
Reviewer 1 Report
Comments and Suggestions for Authors
The author has satisfactorily answered all the questions raised by the reviewers and is recommended for acceptance.
Author Response
We thank the reviewer for the positive acknowledgment. We appreciate the constructive feedback from all reviewers and we are pleased that the responses have addressed their concerns satisfactorily.
Reviewer 2 Report
Comments and Suggestions for Authors
After careful examination of the revised manuscript, the response of the authors to previous reviews, and the changes made in the manuscript, I gather that the revised version of the manuscript has addressed the major concerns raised in the previous version of the paper (Figure 1 should be improved).
Author Response
We thank the reviewer for the constructive feedback. In response, we have made minor but targeted edits to Figure 1 and its legend to more specifically outline the molecular pathways perturbed by the MASLD-associated SNPs. These enhancements aim to clarify key mechanistic details while maintaining the figure’s accessibility and broad appeal, consistent with the review article format. We hope these refinements satisfactorily address the reviewer’s concerns.